# A Possible Accessory Muscle of the Serratus Posterior Superior Muscle

**Kerrie Lashley [1],\* and Guinevere Granite [2]**

[1] Department of Anatomy and Cell Biology, George Washington University, Washington, DC 20037, USA
[2] Department of Surgery, Uniformed Services University of the Health Sciences, Bethesda, MD 20814, USA; guinevere.granite@usuhs.edu
\* Correspondence: klashley81@gwu.edu

**Abstract:** Anatomical variation is defined as the normal range of possibilities in the topography and morphology of body structures. In contrast, an anomaly is any structural or functional anatomical finding beyond the normal range of possibilities. This case study describes a muscular anomaly found in a 73-year-old preserved Caucasian male. We observed a left-sided anomalous muscle originating from the transverse process of the C1 (Atlas) vertebra and inserting onto the proximal attachment of the serratus posterior superior (SPS) muscle at the C7 level. We suggest that this anomaly is a result of early embryological development and hypothesize that the atypical neck muscle may reinforce the action of the SPS. This finding is rare and no reference of it can be found in the literature. Reporting anatomical anomalies is important for the medical literature and education.

**Keywords:** accessory muscle to serratus posterior superior; muscular anomalies; variations in human myology; anatomical variations





## 1. Introduction

Muscular variations are common in the human body and are well-documented in the literature. These variants occur within the normal range of possibilities that may be comprised of morphometric changes, supernumerary elements, consistency and spatial orientation [1,2]. Muscular anomalies are much rarer, and anatomists observe them during routine cadaveric dissection or surgeons discover them inadvertently during surgical procedures [3]. Anomalies are typically asymptomatic and may go unnoticed during the individual's life. With the development of non-invasive imaging techniques and the increase in cadaveric studies, however, anatomic anomalies are being detected more frequently [2].

In this case study, we observed a rare unilateral anomalous muscle in the left posterior neck of a human cadaver. It was located deep to the rhomboid muscles and coursed superior and medial to the serratus posterior superior (SPS) muscle. In addition, we found differences in the cervical origins of the right and left levator scapulae muscles (LSMs), resulting in muscular asymmetry in this individual. This anomalous muscular development is uncommon, and we were unable to find a description of such a case in the literature. Therefore, it is important to provide a detailed description of this rare anomalous muscle.

## 2. Materials and Methods

During cadaveric dissection, we detached the superficial extrinsic back muscles, including the trapezius, latissimus dorsi, and the rhomboid muscles from their proximal attachments and reflected them. The levator scapulae muscle was not reflected. This exposed the intermediate extrinsic back muscles, including the serratus posterior superior (SPS). We detached the SPS from the spinous processes at the C7-T2 vertebral level. We observed the anomalous muscle during the detachment of the SPS at its proximal attachments. Delicate dissection of the anomalous muscle allowed us clear definition and further

examination of its attachment sites. We detached the deep intrinsic muscle, the splenius muscle, proximally from the nuchal ligament and spinous processes of C7-T6 vertebrae. We conducted this dissection bilaterally.

### 3. Case Report

During routine dissection of the neck and back regions, we found a unilateral anomalous muscle occurring within a 73-year-old Caucasian male human cadaver. The anomalous muscle originated from the left transverse process of the atlas and extended inferiorly to insert onto the SPS (Figure 1). The left SPS presented its common anatomy: proximal attachments from the spinous processes of C7-T2 vertebrae and its distal attachments to the superior borders of the second through fifth ribs. The tendinous fibers of the anomalous muscle inserted superficially, and slightly perpendicular onto the left SPS proximal end. The fused fibers of the two muscles attached to the spinous process of the C7 vertebra (Figure 2). The anomalous muscle shared a common attachment at the transverse process of the atlas with the upper portion of the splenius cervicis muscle and two of the upper muscular slips of the left LSM (Figure 2). In length, the anomalous muscle measured 12.5 cm and 0.65 cm wide with a consistent width throughout its muscle belly.

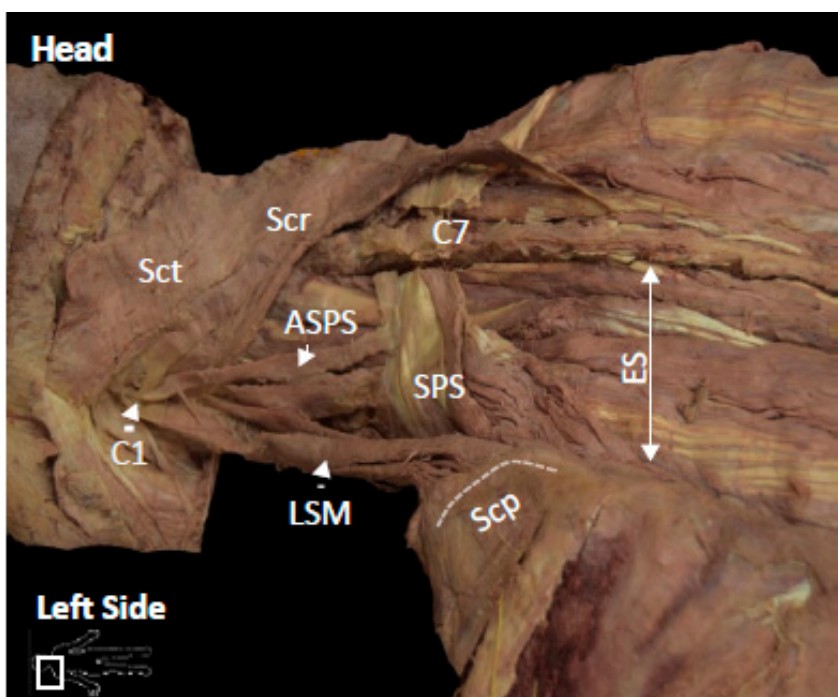

**Figure 1.** Overview of the Posterior Intermediate/Deep Neck. Accessory Serratus Posterior Superior (ASPS) Muscle; Atlas transverse process (C1); Spinous process of cervical vertebra 7 (C7); Erector Spinae (ES) Muscles; Levator Scapulae Muscle (LSM); Scapula (Scp); Serratus Posterior Superior (SPS) Muscle; Splenius capitis (Sct) Muscle; Splenius cervicis (Scr) Muscle.

In combination with this rare finding, there were some noteworthy differences in the cervical origins of the left LSM compared to the right. The left sided LSM exhibited four main muscular slips with distal attachments to the superior angle of the scapula but varied in the number of muscular slips and cervical origins. The observed variabilities of the left LSM are as follows: (1) muscular slips 1 and 2 shared a common cervical origin at C1; (2) the third, broader muscular slip, bifurcated midway, with one muscular slip of cervical origin at C3 and the other at C4; and (3) the fourth muscular slip originated from C5; no LSM muscle slips originated from C2 (Figure 3A). The right-sided LSM had four muscular slips that originated from the transverse processes of C1–C4 vertebrae, respectively, and

were relatively consistent in size and shape (Figure 3B). The muscular asymmetry in the present case produced a five-headed left LSM and a four-headed right LSM.

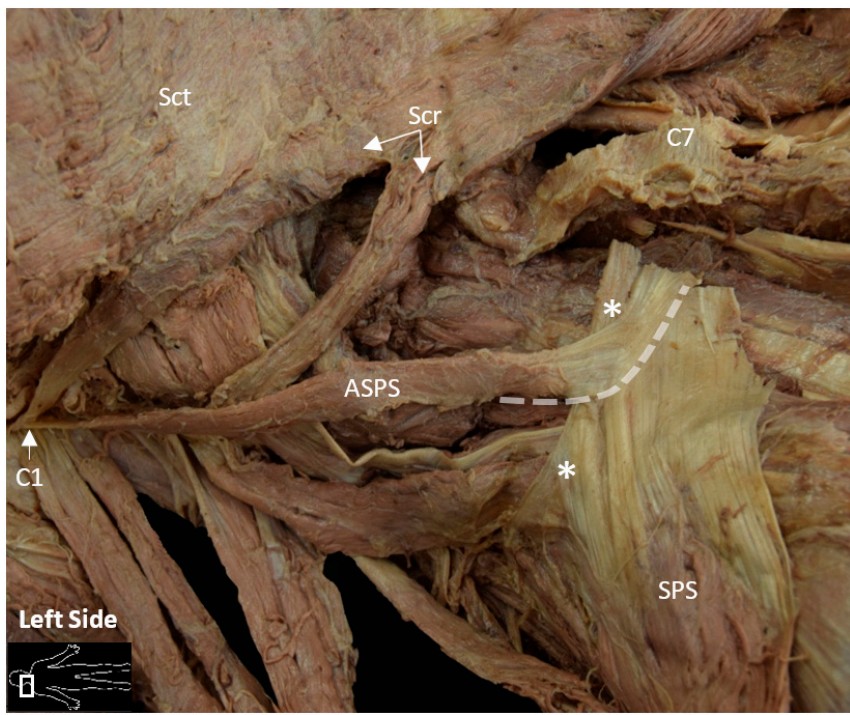

**Figure 2.** Origin and Insertion Sites for ASPS. The dashed line illustrates the slightly perpendicular insertion of the ASPS onto the proximal portion of the SPS. Asterisks indicate the upper tendinous connection of the SPS traveling deep to the ASPS. Accessory Serratus Posterior Superior (ASPS) Muscle; Atlas transverse process (C1); Spinous process of cervical vertebra 7 (C7); Serratus Posterior Superior (SPS) Muscle; Splenius capitis (Sct) Muscle; Splenius Cervicis (Scr) Muscle.

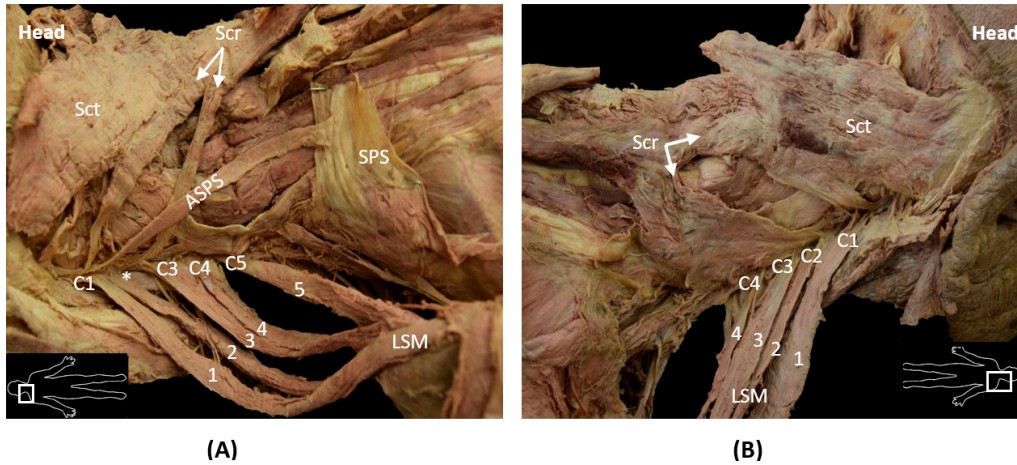

**(A)**                                      **(B)**

**Figure 3.** A and 3B: Bilateral Asymmetry. (**A**) Illustrates left-sided variations in LSM anatomy. Differences in cervical origins of the five muscular slips of the left LSM. Shared vertebral origin of LSM upper slips 1 and 2, distal bifurcation of middle slip producing slips 3 and 4 with different cervical origins, no muscular slip originating from C2 (*). (**B**) Illustrates common anatomy of right sided LSM. Four muscular slips of the LSM with no variation in cervical origins. Accessory Serratus Posterior Superior (ASPS) Muscle; Atlas transverse process (C1); Transverse processes of cervical vertebrae 2–5 (C2-5) Spinous process of cervical vertebra 7(C7); Levator Scapulae Muscle (LSM) which includes the Levator Scapulae Muscle Slips 1–5; Serratus Posterior Superior (SPS) Muscle; Splenius capitis (Sct) Muscle; Splenius Cervicis (Scr) Muscle.

## 4. Discussion

There are earlier reports of supernumerary and divergent muscular slips of the LSM with varying cervical origins and insertions [4]. However, these descriptions always illustrated muscular or tendinous connections to the scapula and were characterized as deviant or rogue slips of the LSM. These deviant slips of the LSM fall within the normal range of possibilities and are true variants of the LSM. In the present case, however, the atypical muscle, Accessory Serratus Posterior Superior (ASPS), does not have any muscular or tendinous connections to the scapula. Its morphology is distinctly different from previous observations of deviant slips of LSM. Therefore, we consider this finding to be a true representation of a rare muscular anomaly and not a deviant or variant slip of the LSM. Furthermore, the insertion of the anomalous muscle onto the dorsal surface of the SPS suggests that it may have aided or reinforced the action of the SPS during elevation of the ribs, specifically the second rib, thus functioning as a possible accessory muscle to SPS. Its insertion does not support its action as an accessory LSM because it cannot elevate or rotate the scapula.

### 4.1. Embryological Implication of the ASPS

Muscle development is a complex and multistep process that involves a series of cellular events of cell fusion, migration and attachment [5]. During myogenesis, myoblasts form into migrating myofibers that are guided towards specific tendon attachment sites to establish connection with the skeletal system [5]. Errors may occur in migration that alter the attachment of muscle and tendon precursor cells, causing ectopic origins and insertions of muscular slips [6]. Therefore, the presence of the ASPS muscle may be a result of errors in the migration of muscle progenitors or tendinous attachment progenitors at the C1 level [7], due to alterations in the microenvironment through which the cells migrate, or Hox-gene related alterations in the axial identity of the progenitors [8].

### 4.2. Muscular Variation Bias to One Side of the Body

While bilateral symmetry in the body's topography is a central feature of vertebrate anatomy [9], many studies demonstrate that asymmetry is not uncommon [10]. In the present case, muscular variation biased toward the left side of the body suggests a disturbance during development may have affected the symmetry of the musculoskeletal system. In a recent study by Bang et al., 2015, multiple muscular variations from the neck to the lower extremities primarily on the left side were reported in a single cadaver [6]. They attributed this lack of symmetry to momentary disturbances in the formation of the embryonic somites.

Somites, which give rise to the vertebrae, ribs and associated muscles and tendons, arise as bilaterally symmetric, paired blocks of paraxial mesoderm in a developing embryo's left and right sides [11]. Each somite pair carried a specific axial identity due to the expression of a unique combination of Hox genes [8]. Interestingly, it has been found in several animal models that asymmetries in left–right somite pairs occur when Retinoic acid (RA) signaling is reduced, leading to developmental delay on one side [11]. These findings suggest that RA signaling is essential to ensure symmetrical mesoderm segmentation and bilateral symmetry along the left–right axis [11]. We speculate that environmental factor(s) acting during early embryonic development may have disturbed this synchrony, leading to a disturbance in the synchronicity of the left and right sides during somitogenesis, causing muscular variation bias in this individual. One such known disruptor of RA signaling is maternal consumption of alcohol [12].

### 4.3. Clinical Significance

Embryonic development of the back and neck muscles is complex. Understanding this process is key to identifying congenital anomalies, diagnosing congenital abnormalities and discerning how and where during development they may occur. Anatomic anomalies and variations of the back and neck may contribute to pathological conditions of these

regions such as myofascial syndrome and the need for radical neck dissection surgery. The presence of the ASPS may pose a challenge to surgeons and could result in erroneous diagnosis as a soft tissue tumor in radiologic evaluations. Hence, reporting is not only important for the medical literature and education, but to also help clarify unknown or controversial clinical presentations in the human body. Moreover, due to a limited medical report, there was no reported medical significance related to the anomaly and asymmetry in this individual nor was the cause of death related to our findings. This case report further supports clinical observations that muscular variations, anomalies and asymmetry can exist simultaneously in an individual and that they may occur more commonly than previously thought.

**Author Contributions:** Conceptualization, K.L.; investigation, K.L.; writing—original draft preparation, K.L.; visualization, K.L.; writing—review and editing, K.L. and G.G. All authors have read and agreed to the published version of the manuscript.

**Funding:** This research received no external funding.

**Institutional Review Board Statement:** Ethical review and approval were waived for this study, given that the material used in this study is considered as a non-human subject; therefore, ethical approval was waived.

**Informed Consent Statement:** Not applicable.

**Data Availability Statement:** Data sharing not applicable.

**Acknowledgments:** With great respect and honor, we thank our donor and his family for such an incredible gift. We thank Ronald W. Rivenburgh, CIV from the Anatomical Gift Program at USUHS, Osvaldo Bustos, Ayo P. Doumatey, Victor Taylor II and Sally Moody for their guidance, mentorship and appreciated contributions. Special thanks to Mohammed Ashraf Aziz for his unwavering support and inspiration.

**Conflicts of Interest:** The authors declare no conflict of interest.

**Disclaimer:** The opinions or assertions contained herein are the private ones of the authors/speakers and not to be construed as official or reflecting the views of the Department of Defense, the Uniformed Services University of the Health Sciences or any other agency of the U.S. Government.

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
