# Peer review of "A Possible Accessory Muscle of the Serratus Posterior Superior Muscle"

_reports, doi:10.3390/reports4010002_

Round 1
Reviewer 1 Report
The authors addressed all the concerns well in the resubmission.
Reviewer 2 Report
The authors found a one sided muscle anomaly in the neck of a preserved 73-year-old Caucasian male cadaver. They suggest this alteration have embryonic origin. The observation is well presented and the case description is sufficiant. They do not exagerate their finding.
Author Response
Please see attachment.

This manuscript is a resubmission of an earlier submission. The following is a list of the peer review reports and author responses from that submission.
Round 1
Reviewer 1 Report
The manuscript entitled “A Possible Accessory Muscle of the Serratus Posterior Superior Muscle” reports a muscular anomaly, which is termed “Accessory Serratus Posterior Superior (ASPS)”, and asymmetry seen in the levator scapulae muscle (LSM). What reported in this manuscript is rare and valuable for understanding muscle development and related clinical significance.
The manuscript is well-written, but there are some issues needed to be addressed:
1) Figure 2 should include C1 position.
2) In figure 2, at where the label “ASPS” is, it seems like there is another tendon attachment of ASPS.
3) Figure 1 and 2 should not use the same legend.
4) From figure 3A, it is possible that ASPS could originally be part of slip 1 of the left LSM. Compared with slip 1 of the right LSM, is left slip 1 smaller?
5) In the discussion, the authors discuss anomaly and asymmetry separately, however, since these two phenomena happen closely in terms of location, they could be resulted from the same reason. This possibility should be discussed.
6) If the anomalies observed happened in embryonic development, there might be some changes in bone. Any findings in bone? In addition, how fetal development could contribute to the development of ASPS?
Minor issue:
1) The title of Figure 3 should be fixed: Figure 3. and 3B.
2) Any medical records could be related with the reported anomaly and asymmetry?
Reviewer 2 Report
The authors found a one sided muscle anomaly in the neck of a preserved 73-year-old Caucasian male cadaver. They suggest this alteration have embryonic origin. The observation is well presented and the case description is sufficiant. They do not exagerate their finding.